# Interplay of Cellular Nrf2/NF-κB Signalling after Plasma Stimulation of Malignant vs. Non-Malignant Dermal Cells

**DOI:** 10.3390/ijms252010967

**Published:** 2024-10-11

**Authors:** Kristina Manzhula, Alexander Rebl, Kai Budde-Sagert, Henrike Rebl

**Affiliations:** 1Institute of Cell Biology, Rostock University Medical Center, 18057 Rostock, Germany; kristina.manzhula@med.uni-rostock.de; 2Research Institute for Farm Animal Biology (FBN), Wilhelm-Stahl-Allee 2, 18196 Dummerstorf, Germany; rebl@fbn-dummerstorf.de; 3Institute of Communications Engineering, University of Rostock, 18051 Rostock, Germany; kai.budde-sagert@uni-rostock.de

**Keywords:** dermal cells, cold atmospheric plasma, oxidative stress, NF-κB, Nrf2, translocation

## Abstract

Skin cancer is one of the most common malignancies worldwide. Cold atmospheric pressure Plasma (CAP) is increasingly successful in skin cancer therapy, but further research is needed to understand its selective effects on cancer cells at the molecular level. In this study, A431 (squamous cell carcinoma) and HaCaT (non-malignant) cells cultured under identical conditions revealed similar ROS levels but significantly higher antioxidant levels in unstimulated A431 cells, indicating a higher metabolic turnover typical of tumour cells. HaCaT cells, in contrast, showed increased antioxidant levels upon CAP stimulation, reflecting a robust redox adaptation. Specifically, proteins involved in antioxidant pathways, including NF-κB, IκBα, Nrf2, Keap1, IKK, and pIKK, were quantified, and their translocation level upon stimulation was evaluated. CAP treatment significantly elevated Nrf2 nuclear translocation in non-malignant HaCaT cells, indicating a strong protection against oxidative stress, while selectively inducing NF-κB activation in A431 cells, potentially leading to apoptosis. The expression of pro-inflammatory genes like *IL-1B*, *IL-6*, and *CXCL8* was downregulated in A431 cells upon CAP treatment. Notably, CAP enhanced the expression of antioxidant response genes *HMOX1* and *GPX1* in non-malignant cells. The differential response between HaCaT and A431 cells underscores the varied antioxidative capacities, contributing to their distinct molecular responses to CAP-induced oxidative stress.

## 1. Introduction

Skin cancer, also known as cutaneous carcinoma, is the most common malignant disease worldwide. Its incidence is increasing by approximately 5% annually, affecting over two million people each year [1,2], and it is expected to further increase due to the thinned ozone layer, exposure to ultraviolet radiation, and an ageing population. Although existing treatments are generally effective, they can cause considerable pain and disfigurement, which has a negative impact on patients’ quality of life. Therefore, minimising adverse effects and reducing relapse and mortality rates remain a major concern [3]. Thus, developing innovative and targeted treatment options is crucial [4]. In this regard, cold atmospheric pressure plasma (CAP) is proving to be a promising tool for skin cancer therapy [5,6,7,8,9]. Some skin cancers, particularly advanced melanomas, can become resistant to conventional treatments like chemotherapy or targeted therapies [10]. Cold plasma offers a different mechanism of action that might help overcome drug resistance, providing an alternative or supplementary option for difficult-to-treat cancers [11,12,13]. CAP is a partially ionised gas with a temperature of about 37 °C. It is produced by supplying energy to a neutral gas and is therefore also referred to as the fourth state of matter [14]. In addition to free charged particles, CAP generates other physical and chemical variables such as UV photons (100–380 nm), electromagnetic fields, heat, and biologically active species, including radicals [15,16]. CAP has been shown to be effective in anti-septic and anti-inflammatory applications as well as in promoting cell proliferation and tissue regeneration [17,18,19] due to the generation of reactive oxygen and nitrogen species (RONS). On the other hand, anti-proliferative and anti-tumour effects have also been observed with prolonged exposure, suggesting a dose-dependent effect [5,20,21]. More precisely, several studies have shown efficient partial remissions after treatment of various cancers such as breast and bladder cancer, as well as malignant melanoma in vitro and head and neck cancer in vivo [5,22,23]. Despite many years of research, much remains unclear about the exact biological effects of CAP and the underlying mechanisms [6]. Therefore, further investigation of its selective effect on cancer cells at the molecular level is essential before the practical application of CAP is possible.

Cells are constantly exposed to oxidative stress, a condition characterised by impaired redox homeostasis due to increased production of reactive oxygen species (ROS) or decreased ROS scavenging capacity [24]. ROS are naturally generated as by-products of numerous cellular mechanisms, including mitochondrial respiration, and regulate cellular redox balance [25,26,27]. Since ROS are highly reactive with biological molecules, a large increase could lead to oxidation of intracellular proteins, lipids, or DNA and alter their function, which in turn could trigger apoptosis [28,29]. To counteract oxidative stress and its consequences, cells are equipped with an antioxidant system that includes ROS-scavenging enzymes, such as glutathione peroxidase/reductase (GPx, GR), superoxide dismutase (SOD), and catalase (CAT), as well as small molecules such as glutathione (GSH) and reduced nicotinamide adenine dinucleotide phosphate (NADPH) [26,30]. Interestingly, the cellular characteristics related to sensitivity to oxidative stress differ in cancer cells from those of their non-malignant counterparts [31].

Higher basal levels of ROS and altered levels of ROS-scavenging enzymes have been observed in cancer cells in vitro and in vivo [31,32]. Increased ROS formation can trigger a compensatory redox adaptation response to keep ROS levels just below the apoptotic threshold. However, cancer cells exceed the apoptotic threshold faster when exposed to additional ROS stress than non-malignant cells, which have a “spare” antioxidant capacity. This difference is considered a key aspect in the action of CAP [25]. Beyond their elevated endogenous ROS production, cancer cells possess additional attributes, rendering them more receptive to exogenous ROS. For once, the transmembrane diffusion of RONS is proposed to be easier in cancer cells due to an elevated expression of aquaporins, which facilitate transmembrane diffusion of H2O2 [33]. Moreover, cancer cells often exhibit diminished levels of cholesterol, a critical membrane lipid protecting against RONS. Consequently, free radicals induce increased lipid peroxidation, forming membrane pores of sufficient sizes (~15 Å) to enable the diffusion of free reactive species [34]. Additionally, CAP treatment is associated with damage to the antioxidant system, as decreased NADPH/NADP+ and GSH/GSSG ratios have been observed, as well as decreased expression of SOD and CAT, further augmenting the existing redox imbalance in cancer cells [35,36]. 

CAP’s central anti-tumour effect is presumably based on apoptosis, a physiological form of programmed cell death, in which the cell actively engages in self-destruction [37,38]. Consequently, the study of associated signalling pathways, of nuclear factor-erythroid 2-related factor 2 (Nrf2) and nuclear factor kappa B (NF-κB), could provide valuable insights into the mechanisms underlying the differential effects of CAP in normal and tumour cells [39,40].

Nrf2—the master regulator of oxidative stress responses—was first isolated and described in 1994 by Moi et al. [41] as an activator of β-globin gene expression and is now recognised as the main activator of mechanisms that scavenge excessive ROS levels and restore redox homeostasis [41,42]. Under basal conditions, two Kelch-like ECH-associated protein 1 (Keap1) molecules homodimerise and bind Nrf2, sequestering it in the cytoplasm (Figure 1A). The ubiquitin ligase Cullin 3 associated with Keap1 ubiquitinates lysine residues of Nrf2 for its subsequent proteasomal degradation while Keap1 is regenerated. 

However, in the case of oxidative stress, the cysteine thiols of Keap1 react with various oxidative and electrophilic molecules, leading to conformational changes of the Keap1 molecule and disruption of the interactions between Nrf2 and Keap1. As a result, Nrf2 is released and translocates to the nucleus (Figure 1A), where it binds to the antioxidant-responsive element (ARE) present in the promoters of numerous cytoprotective genes. These encode stress response proteins and metabolic enzymes such as antioxidant phase II proteins (haeme oxygenase-1 (*HMOX-1*), NAD(P)H quinone dehydrogenase (*NQO1*), and *CAT*) [43,44,45].

Research has revealed the ubiquitous expression of NF-κB —a key transcription factor regulating cellular and immunological processes—in different cell types and its central role in the control of various signalling pathways, including the regulation of metabolic and immunological processes as well as cell proliferation, differentiation, and apoptosis. Thus, NF-κB plays a crucial role in the complex process of cancer development [46,47,48]. NF-κB exhibits a broad spectrum of signal-, context-, and cell-type-specific effects since a number of stimuli activate different NF-κB subunits, resulting in diverse cellular responses. The most important stimuli include bacterial lipopolysaccharide, tumour necrosis factor (TNFα), interleukin (IL)-1β, as well as ribotoxic, genotoxic, oxidative, and shear stress [39,49]. Upon activation, NF-κB dimers are released from prior association with members of the inhibitor of kappa B (IκB) family to translocate to the nucleus (Figure 1B), bind κB-containing DNA on a large scale, and control broad gene expression programmes [39,50]. This rapid and transient translocation of NF-κB within a few minutes is not dependent on protein synthesis [51,52]. To prevent excessive NF-κB activity in response to transient inflammatory signals and to allow reactivation in case of persistent IKK activity, IκBα is resynthesised to inhibit NF-κB activity. Therefore, IκBα translocates into the nucleus, binds NF-κB, and transports it back into the cytosol [39]. This negative feedback loop of IκBα is just one example of a complex regulatory network that controls NF-κB signalling [53,54]. Overall, numerous genes respond to NF-κB, including those encoding cytokines (e.g., *IL-1*, *IL-6*, chemokine (C-X-C motif) ligand (*CXCL8*), transforming growth factor (*TGFβ*), and *TNFα*), as well as critical proteins such as cell adhesion molecules, anti-apoptotic proteins, matrix metalloproteinases, and other enzymes. This comprehensive gene regulation emphasises the importance of NF-κB as a master regulator of overall cell survival [39].

This article aims to shed light on the diverging dose-dependent anti-tumour effect of CAP on squamous cell carcinoma cells (SCC, A431) in contrast to their non-malignant counterparts (HaCaT). With that objective, this study focused in particular on investigating the efficacy of CAP on cells under identical culture conditions, as the non-comparability of the culture media composition was a major drawback of past studies. Furthermore, basal ROS and antioxidant levels were compared in both cell lines. In addition, the interplay between Nrf2 and NF-κB signalling pathways in the effect of CAP was investigated as a function of the dose. This included observing and quantifying the translocation dynamics of the two transcription factors and their respective inhibitors as well as measuring the protein abundance of the associated molecules.

## 2. Results

### 2.1. Characterisation of Unstimulated Malignant vs. Non-Malignant Dermal Cells

Basal intracellular ROS and antioxidants, as well as their correlation with relevant protein levels, were assessed in malignant A431 and non-malignant HaCaT cells. The basal ROS levels (Figure 2A) of A431 and HaCaT cells appeared comparable. In contrast, antioxidant levels in unstimulated cells were significantly higher (1.4-fold) in A431 cells (Figure 2B), indicating a generally higher metabolic turnover as expected for tumour cells. Also, basal levels of proteins involved in the cytosolic antioxidant sensing pathways were quantified, i.e., NF-κB, IκBα, Nrf2, Keap1, as well as IKK and pIKK (Figure 2D). Overall, the quantification revealed similar values in both cell lines. However, Keap1 (+42%) and IKK (+82%) exhibited a significantly higher basal abundance in HaCaT cells, suggesting an increased basal Nrf2 inhibition. Thus, a higher Keap1 abundance might indicate the “spare” redox adaptation capacity in HaCaT cells. Regarding the higher IKK levels, it can be hypothesised that HaCaT cells can respond more rapidly and robustly to NF-κB stimuli, as an increased IKK availability would allow for more rapid phosphorylation/activation, thereby triggering the release of NF-κB.

The basal nucleus-to-cytosol ratio of the key players in these pathways was identical in both unstimulated A431 and HaCaT cells. Conversely, the ratio of IκBα was slightly higher in A431 (+15%) compared to HaCaT cells. This suggests a higher nuclear IκBα localisation, potentially inhibiting pro-apoptotic NF-κB activity and supporting cell survival in malignant A431 cells. The basal Keap1 nucleus/cytosolic localisation was comparable in both cell lines, with a slight trend towards higher nuclear levels in A431 cells (+8%). Similarly, A431 cells displayed a slightly higher basal Nrf2 nuclear localisation (+9%) relative to HaCaT cells (Figure 2C) and thus an increased level of activated Nrf2. This may indicate a cellular adaptation to elevated basal ROS levels, commonly observed in various cancer types. This basal redox adaptation could explain the comparable ROS levels measured in both cell lines, as A431 cells upregulate their antioxidative system in order to keep oxidation and reduction events in balance, thus resisting the change in intracellular oxidative stress in a basal state. This notion is supported by the higher antioxidant levels in unstimulated A431 cells (Figure 2B). After characterising the differences in basal redox status, we now investigate how these variations affect cellular responses to CAP-induced oxidative stress.

### 2.2. Comparison of CAP Effect between Dermal Tumour and Non-Malignant Cells

Several studies have demonstrated a selective inhibitory effect of CAP on tumour cells [9,18]. To verify this in the context of the utilised cell lines and plasma setup of this study, the cell viability was assessed after exposure to plasma. We found that post CAP treatment, the metabolic activity of tumourigenic A431 cells significantly decreased with an escalating exposure time (Figure 2E). In contrast, HaCaT cells exhibited a comparatively slower decline in metabolic activity. 

Furthermore, this study aimed to investigate how this selective effect on the cells manifests over time and whether it occurs even after short exposure times. Measurements of the oxygen consumption rate over time (Figure 3) revealed that cellular respiration in A431 cells was significantly inhibited by a 60 s plasma treatment (significant from time point tA = 5.5 h). HaCaT cells displayed a milder downward trend in response to 60 s treatment. When comparing a treatment time of 90 s, both cell lines show significant inhibition of the respiration rate, whereby the oxygen consumption of A431 cells was more strongly influenced. Upon H_2_O_2_ treatment, A431 displayed a persistent downward trend, whereas HaCaT showed a steady respiration rate with a slight recovery over time. Building on the observed differences in cellular respiration following CAP and H_2_O_2_ treatment, we next examine the role of RONS, which may contribute to the differential oxidative stress responses observed between the cell lines.

### 2.3. Differences in Antioxidant Levels

Measurements of the hydrogen peroxide concentration in CAP-treated medium revealed that with an increasing CAP treatment time, significantly more hydrogen peroxide was detectable in a dose-dependent manner (Figure 4A). Thus, a progressive overload of the cellular antioxidative system seems plausible as a consequence of CAP treatment.

The strain on the antioxidant system becomes apparent when observing the significant decrease in antioxidant capacity following CAP stimulation in A431 cells (Figure 4B). In contrast, plasma stimulation in non-malignant HaCaT cells resulted in an increase in intracellular antioxidant levels in response to oxidative stress, facilitating a robust redox adaption.

### 2.4. Translocation of Nrf2/Keap1 upon CAP Treatment 

To further examine the differential redox adaptation as a response to CAP in cancer and non-malignant cells at a molecular level, key players in the ROS sensing machinery were analysed. Since the activation of these signalling pathways is based on an altered localisation within the cell compartments, the translocation of the proteins was quantified based on high-resolution microscopy (for exemplary images, see Appendix A). 

The non-tumourigenic HaCaT cells exhibited a steady and significantly elevated Nrf2 nuclear translocation after plasma treatment, suggesting a robust antioxidant reaction and supporting the observed increase in antioxidant capacity. The Nrf2 localisation in A431 cells only showed a minor upward trend in nuclear translocation with increasing durations of CAP treatment (Figure 5A), although measurements of protein levels revealed an elevated Nrf2 abundance in CAP-treated A431 cells (+48%) (Figure 6A). This indicates that only a slight improvement in redox adaptation is feasible, as the adaptation capacity seems to be depleted. As anticipated, the nucleus-to-cytosol ratio of Keap1 did not change upon CAP treatment in both cell lines (Figure 5B). However, the protein measurements revealed a significantly elevated Keap1 abundance in HaCaT cells (+54%) after CAP exposure, whereas IKKβ expression was slightly downregulated (−23%) (Figure 6). Interestingly, A431 cells showed significantly divergent IKKβ expression relative to their non-malignant counterparts (+25%). However, Keap1 abundance was heightened after CAP treatment (+81%), similar to HaCaT cells (+54%).

### 2.5. Translocation of NF-κB/IκBα upon CAP Treatment

Given NF-κB’s redox sensitivity and association with apoptosis, NF-κB, alongside its inhibitor IκBα, were analysed as well. Indeed, A431 cells (Figure 5C) gradually increased NF-κB translocation with an escalating CAP exposure time. Hence, after 180 s of CAP treatment, a significantly elevated translocation (+20%) was observed. In contrast, HaCaT cells did not show a CAP-dose-dependent effect on NF-κB translocation. Regarding the nucleus-to-cytosol ratio of IκBα after CAP stimulation, a general decline was evident in A431 cells, gradually increasing with an escalating exposure time, ultimately converging to the control ratio of 1 after 180 s. Accordingly, this implies severely reduced NF-κB inhibition in the nucleus. In HaCaT cells, a significant elevation in IκBα translocation was noted after 120 (+18%), 150 (+21%), and 180 s (+30%). However, the quantitative analysis indicated that this effect was based on the decrease in IκBα in the cytosol rather than on a nuclear IκBα accumulation (e.g., 180 s: nucleus; ±0%, cytosol: −30%). Indeed, the protein quantification displayed a decrease in IκBα abundance after 180 s of CAP treatment in HaCaT cells (−20%) (Figure 6), further supporting this observation.

**Figure 6 ijms-25-10967-f006:**
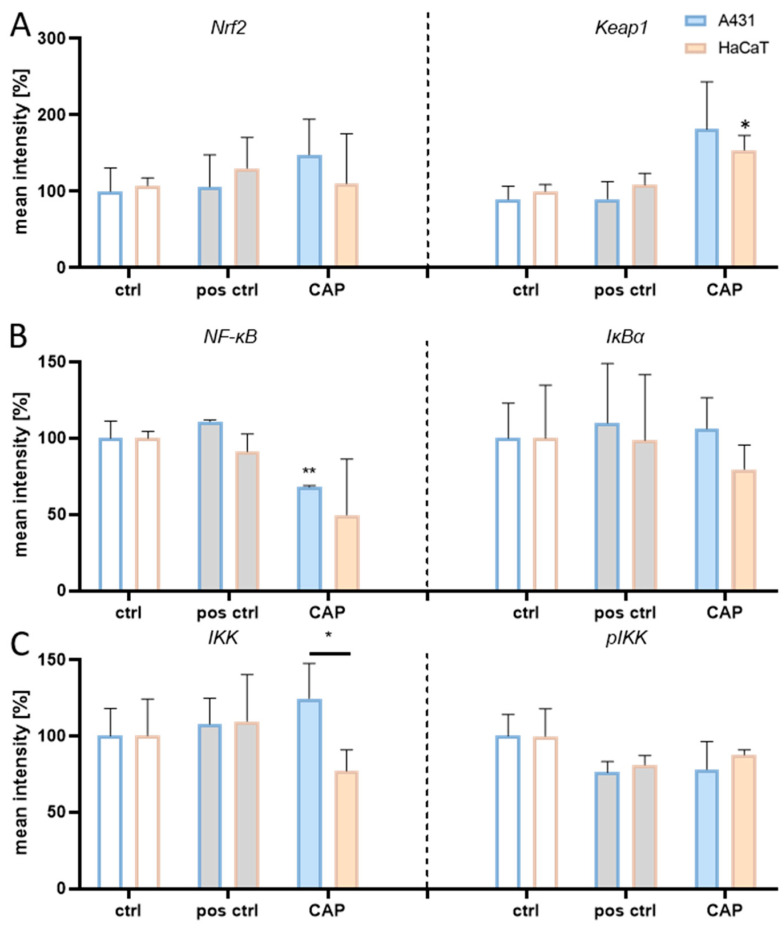
Mean intensities of proteins associated with Nrf2 and NF-κB signalling in A431 and HaCaT cells. (**A**, right) In HaCaT cells, the Keap1 abundance was significantly increased after 180 s exposure to cold atmospheric pressure plasma (CAP). (**B**, left) Also, A431 cells exhibited significantly reduced levels of NF-κB after CAP treatment. (**C**, left) The abundance of IKK was significantly reduced in HaCaT cells relative to A431 cells. For positive controls (grey filling), 15 µg/mL H_2_O_2_ and 20 μg/mL Poly(I:C) were used, respectively (* *p* ≤ 0.05; ** *p* ≤ 0.01, n = 3).

Furthermore, quantification of NF-κB abundance revealed significantly diminished protein levels in A431 cells after CAP exposure (Figure 6B). Accordingly, quantitative analyses of NF-κB’s total nuclear/cytosolic localisation showed diminishing NF-κB levels in CAP-exposed A431 cells. Overall, these findings indicate that CAP treatment selectively and dose-dependently activates NF-κB signalling in A431 cells via the canonical pathway, resulting in apoptosis induction. Conversely, NF-κB signalling is not activated in HaCaT cells, as the cells successfully adapt to the additional ROS generated by CAP, thus not crossing the apoptotic threshold. Having examined the translocation dynamics of these pathways, we elucidated whether subsequent target genes are activated.

### 2.6. Gene Expression

To investigate the effects downstream of the observed translocation of the Nrf2/Keap1 and NF-κB/IκBα systems, the expression of selected target genes was analysed post CAP treatment. Overall, we found that CAP treatment resulted in a downregulation of pro-inflammatory genes in the tumourigenic A431 cells compared to their non-malignant counterparts (Figure 7A). CAP treatment downregulated the transcript numbers (Figure 7B) of *IL-1B*, *IL-6,* and *CXCL8* in A431 cells compared to HaCaT cells. *TNFα*, which has quite low abundances in these cell lines, there was an increase in expression, although this was less pronounced in A431 cells. The expression of *CALR*, a marker for immunogenic cell death, remained unchanged.

The proliferation-promoting factor transforming growth factor beta 1 (*TGFB1*) was downregulated upon treatment. This indirect proliferation-inhibiting effect of plasma treatment was more pronounced in tumour cells. Similarly, the expression of *SLC7A11*, a cysteine in/glutamate out antiporter, was reduced after CAP treatment, while hydrogen peroxide treatment resulted in increased expression. In contrast, we found a significantly higher expression of the antioxidative response gene *HMOX1* and to a lesser extent, *GPX1* in non-malignant HaCaT compared to A431 cells.

A principal component analysis of the expression profiles in control and treated A431 and HaCaT cells, revealed distinct patterns (Figure 7C). Strikingly, the analysis clustered the individual treatments well, including the three incubation intervals of the plasma treatments. Along dimension 1, the loadings for all genes were negative in A431 cells, while loadings for four out of nine genes analysed (*GPX*, *CALR*, *TGFB1,* and *CAT*) were positive in HaCaT cells. Along dimension 2, only the loadings for *HMOX1* and *SLC4A11* were negative in HaCaT cells, but positive in A431 (Figure 7C). In the non-malignant HaCaT cells, the clusters of untreated control and argon treatment were almost congruent. The cluster of the 60 s plasma treatment was localised towards that of the hydrogen peroxide stimulation, which implies a similar adaptive response to oxidative stress. Indeed, quantification of hydrogen peroxide in PAM showed that the hydrogen peroxide amount that was generated by 60 s of CAP treatment (492 µM) nearly equalled the hydrogen peroxide concentration used as a positive control in all experiments (441 µM) (Figure 4A).

## 3. Discussion

The general ability of CAP to exert a more substantial effect on tumourigenic compared to non-malignant cells has already been described for many cell types in the literature. For that reason, this study focused, in particular, on investigating the efficacy of CAP on malignant and non-malignant cells under identical culture conditions, as the non-comparability of the culture media composition was a major drawback of past studies. Nonetheless, our findings align with earlier studies that have elucidated CAP’s potential in targeting cancer cells. Zirnheld et al. [55] reported on CAP’s ability to selectively eliminate melanoma cells while sparing non-malignant keratinocytes. This great advantage, compared to most conventional cancer treatments, displayed great potential for using CAP in oncotherapy, thus driving further investigations. Up to now, the anti-tumoural effect of CAP has been studied extensively and has shown a significant effect in over 20 different types of cancer cell lines in vitro, including leukaemia, breast cancer, colorectal cancer, lung cancer, and SCC [56,57,58,59,60,61]. Pasqual-Melo et al. [62] reinforced the dose-dependent decline in metabolic activity of A431 and HaCaT cells 24 h after CAP exposure. Also, the treatment was significantly more cytotoxic in A431 cells, aligning with the results obtained in this work. Accordingly, Yan et al. [33], who reviewed the existing literature comparing cancer cells to normal cells from the same tissue type, classified CAP’s effect on SCC as strongly selective. 

### 3.1. Diverging Antioxidative Capacity in Malignant vs. Non-Malignant Cells

It became increasingly accepted that the divergent responses of cancer cells and their non-malignant counterparts to CAP could be attributed to differences in their basal ROS levels [31,37]. However, measured ROS levels appeared comparable between malignant A431 and non-malignant HaCaT cells (Figure 2A). In that context, it is essential to acknowledge that basal ROS levels reflect a complex balance between ROS production and control. Since cancer cells usually have higher ROS levels due to oncogenic stimulation, increased metabolism, and mitochondrial dysfunction, the antioxidant defence systems also play a crucial role in basal ROS levels [30,63]. Indeed, the increased basal antioxidant level (Figure 2B) of the unstressed tumour cells suggests that they already invest significantly more effort in the increased production of antioxidants in order to keep the intracellular ROS level at a stable cell-compatible level (homeostasis). This notion finds support in the slightly elevated basal Nrf2 localisation in A431 nuclei, indicating heightened Nrf2 activity associated with basal redox adaptation (Figure 2C). Nearly all cancer cells show imbalances in antioxidant enzyme levels compared to non-malignant cells [31]. Regarding skin cancer, interesting disparities emerge in the expression profiles of key antioxidant enzymes, such as copper–zinc SOD, manganese SOD, and CAT, as well as the lipid peroxidation product marker protein-bound malondialdehyde. Human melanoma biopsies show overexpressed antioxidant enzymes and elevated malondialdehyde levels, indicating a basal redox adaptation to elevated ROS levels [64]. These insights highlight the fundamental variations between cancer cells and their non-malignant counterparts.

A well-established theory based on disparities in basal ROS levels between malignant cells and their non-malignant counterparts proposes that CAP-generated ROS overwhelm the extensively preloaded antioxidant system in cancer cells, pushing them beyond the apoptotic threshold quicker than normal cells [25,65]. Indeed, we demonstrated that with an escalating CAP exposure time, the hydrogen peroxide, nitrite, and nitrate concentrations significantly and gradually increased (Figure 4A,C), while the antioxidative capacity of malignant A431 decreased in the same fashion (Figure 4B). On the contrary, intracellular antioxidant levels of non-malignant HaCaT cells significantly increased. Regarding the significantly steeper decrease in viability of A431 cells compared to HaCaT cells (Figure 2E), this notion seems plausible. Based on the assessed oxygen consumption rates, which showed a significant inhibition in the cellular respiration of A431 cells after 5.5 h of 60 s CAP exposure (Figure 3), not solely the external but also the internal ROS levels are likely to increase upon CAP exposure,. These experiments are consistent with studies on epithelial cells, in which cell respiration was blocked immediately after the addition of plasma-activated medium (PAM). An immediate significant reduction was observed from 3.5 h of incubation, while adhesion was only affected later [17,66]. With this measurement setup, we could not examine the earliest period of stimulation in the current study, but we suspect a similar direct effect on respiration, as the values of the 60 s treatment showed a 50% reduction in respiration in A431 cells from the start of measurement. The response to single H_2_O_2_ treatment also revealed a strong inhibitory effect on respiration in both cells, although the curve was clearly different from that with CAP, which is also attributable to the presence of other RONS in CAP (Figure 4C). While the respiration rate in A431 steadily decreased, HaCaT showed a stable respiration rate that only slightly increased over time, suggesting cell recovery through an adaptive antioxidant response. This aspect will be addressed in more detail in the following sections. 

### 3.2. Nrf2/Keap1 in CAP-Induced Oxidative Stress Defence

The aforementioned concept of redox adaptation introduced by Kong et al. [65] further elaborates that while the strengthening of defence mechanisms manages to maintain a balance between the harmful effects of ROS and the protective role of antioxidants at a basic level, it fails to react to further increasing ROS levels, especially compared to non-malignant cells. Indeed, the translocation analyses of the Nrf2/Keap1 system showed enhanced Nrf2 activity in non-malignant HaCaT cells (Figure 5A), associated with a successful redox adaptation. Conversely, A431 did not show such adaptation. Thus, we assume that the basal rate of Nrf2 translocation assessed in this study, along with the consequential activation of antioxidant genes, has nearly reached its limit in A431 cells. This circumstance might only allow for a marginal enhancement in response to the additional exogenous oxidative stress induced by CAP. Non-malignant cells seem to possess a more pronounced antioxidative reserve capacity before approaching their threshold, as evidenced by HaCaT’s significantly heightened antioxidant capacity after CAP stimulation relative to A431 cells (Figure 4B). 

Interestingly, multiple mechanisms, e.g., mutations in the Nrf2/Keap1 pathway, cause Nrf2 dysregulation in many cancer types. Thus, heightened basal Nrf2 activation in cancer cells is frequently observed, facilitating tumour progression and resistance to therapy [67]. For instance, the efficacy of ROS-inducing therapies is reduced due to the enhanced basal antioxidative capacity, as illustrated in this work (Figure 4B). Additional mechanisms by which Nrf2 enhances the resilience of cancer cells against therapy include the upregulation of genes involved in drug metabolism as well as drug transport proteins associated with increased detoxification of chemotherapeutic agents, along with those used in radiotherapy and immunotherapy [68]. A recent study further suggests Nrf2’s role in enhanced repair of double-strand breaks, which are the primary effect of radiation therapy [69]. In sight of this, Nrf2-inhibiting and Keap1-enhacing agents are being investigated to overcome this obstacle in cancer therapy [70]. According to the results of this work, a combinatorial therapy with CAP appears particularly promising as it induces a ROS overload that cancer cells cannot effectively adapt to, ultimately triggering cell death.

For HaCaT cells and NHEK primary keratinocytes, such Nrf2 translocation was demonstrated after 15 min of incubation in PAM (60 s) and 50 μM hydrogen peroxide, respectively [71]. Nevertheless, examinations focusing on the translocation pattern of Nrf2 in A431 cells or other SCC cells upon CAP treatment have not been conducted before. However, Bekeschus et al. [72] revealed unaltered Nrf2 localisation in B16F10 melanoma cells after plasma treatment, aligning with the outcomes for A431 cells obtained in this work. 

Similarly, previous studies did not register noteworthy alterations in Keap1 translocation within HaCaT cells after cultivation in PAM (60 s) equivalent to a hydrogen peroxide concentration of ~32 μM [71]. Based on the obtained results, it appears that the intracellular distribution of Keap1 remains relatively unaffected by stimulation with plasma (Figure 5B). Sun et al. [73] questioned the previously accepted cytosolic location of Keap1 and proved its constant shuttling between the nucleus and cytoplasm under basal conditions. Consequently, it was established that Keap1′s role extends beyond mere cytoplasmic Nrf2 sequestration, revealing its function as a post-induction repressor of Nrf2. Once translocated into the nucleus, Keap1 binds to Nrf2 and dissociates it from ARE. Keap1 then facilitates the export of the Nrf2:Keap1 complex into the cytosol, where Nrf2 is subsequently degraded, resulting in the termination of Nrf2 signalling [74,75]. Although the translocation of Keap1, mediated by importin α7 and the C-terminal Kelch domain, is Nrf2-independent, its response to oxidative stress remains elusive [73,74]. Furthermore, the persistence of nuclear Nrf2 accumulation in HaCaT cells observed 1 h after CAP treatment, coupled with Nrf2’s relatively short half-life of 7–18 min, indicates that the redox balance was not restored at this time point [76,77,78]. Thus, Nrf2 inhibition via Keap1 translocation is not expected. 

### 3.3. The Role of NF-κB in Response to CAP Treatment

Ultimately, the inability of cancer cells to adapt to additional ROS as inflicted by CAP leads to ROS overloading and the accelerated surpassing of the threshold, resulting in apoptosis of A431 cells. CAP’s efficiency to induce apoptosis in malignant cells is particularly noteworthy since the evasion of cell death, a fundamental hallmark of cancer cells, represents a great challenge in cancer therapy [79,80]. Besides NF-κB being a key TF regulating cell fate, the intriguing connection between CAP and NF-κB becomes apparent regarding NF-κB’s redox sensitivity [81]. Given the plasma-induced ROS elevation in the media (Figure 4A), we assumed NF-κB has involvement in CAP’s mechanisms. Indeed, we demonstrated a gradual elevation in NF-κB activity with an increasing exposure time to CAP in A431 cells whereas HaCaT cells did not display any changes (Figure 5C). Given the generally accepted anti-apoptotic nature of NF-κB, these results initially seem controversial. However, growing evidence supports a pro-apoptotic role of NF-κB, suggesting its dual stimulus-dependent role in regulating cell death [46]. For instance, TNFα-induced NF-κB activation exerts a cytoprotective, and hence anti-apoptotic, effect, whereas hydrogen-peroxide-induced NF-κB translocation triggers apoptosis in HeLa cells [81]. Yan et al. [82] demonstrated that hepatocellular carcinoma cells (HepG2) exhibited a decreased Bcl-2/Bax ratio after CAP treatment, inducing apoptosis. 

This dual role—facilitating both cell survival and death—presents valuable therapeutic potential alongside inherent risks. Traditionally associated with inflammation and cell survival, NF-κB has a key role in therapy-resistant cancer. Thus, inhibitors of NF-κB emerge as promising strategies to enhance the sensitivity of cancer cells to chemotherapy and radiotherapy by promoting apoptosis. NF-κB inhibition could boost anticancer effects, since NF-κB upregulation plays a crucial role in acquired resistance to ROS-inducing therapies [10]. For example, NF-κB inhibition sensitises cells to radiotherapy-induced apoptosis via JNK pathway activation [83]. Resistance to ROS-elevating therapies, such as chemotherapy and radiotherapy, is also linked to an increased antioxidant capacity in cancer cells. Recent preclinical studies have shown that inhibiting GSH or Thioredoxin (Trx) antioxidant systems, which are downstream of Nrf2 signalling, can sensitise various tumour cell types to cancer therapy. Auranofin, for example, an inhibitor of the thioredoxin antioxidant system, reduces NF-κB expression and overcomes therapy resistance [84]. Similar outcomes were observed with the inhibition of GSH metabolism [85]. However, recent findings indicate that NF-κB activation can lead to apoptosis in tumour cells while sparing healthy cells, highlighting its selective cytotoxicity [86,87]. In this context, inhibition of NF-κB would not be beneficial. Also, it is important to acknowledge that due to the complex function of NF-κB, its inhibition may inadvertently compromise normal immune responses or exacerbate tumour progression through inflammatory mechanisms.

It becomes apparent that NF-κB’s role is context-dependent, influenced by cell type and microenvironmental factors. Understanding the specific molecular pathways governing NF-κB’s differential roles is essential to maximise efficacy while minimising adverse effects of therapeutic interventions. Future research should focus on delineating the conditions under which NF-κB promotes tumour cell death versus survival as well as characterise the role of NF-κB in respective cancer types. This knowledge would enable us to specifically target NF-κB activity according to its respective role. 

Given IKKβ’s role as the activator of NF-κB signalling, and NF-κB’s key role in cancer, it is plausible that the regulation of IKKβ differs significantly between malignant and non-malignant cells [88]. In various cancer types, IKK overexpression has been associated with tumour progression and therapy resistance [89,90]. Also, this overexpression was reported to cause a persistent NF-κB activation [91]. Indeed, our findings showed that IKKβ expression was significantly elevated in A431 tumour cells upon CAP treatment compared to HaCaT cells (Figure 6C, left). Also, NF-κB translocation was enhanced in A431 cells after prolonged CAP exposure (Figure 5). Thus, the results of this work align with the current understanding of IKKß’s role in cancer biology.

In view of the sustained elevated basal Nrf2 translocation in A431 cells, increasing CAP exposure likely overwhelms their ability to handle oxidative stress, leading to apoptosis. This may involve increased NF-κB translocation, enabling upregulation of pro-apoptotic proteins. Considering the contradicting translocation pattern observed in HaCaT cells, this is further supported by the lower viability of A431 cells compared to HaCaT cells subsequent to CAP exposure (Figure 2E). In that context, the significantly diminished NF-κB protein levels in A431 cells after CAP exposure (Figure 6) seem conflicting. Indeed, NF-κB regulation via degradation is relatively uncommon. However, it has been demonstrated that p65 is cleaved by caspase-3 at its N-terminus (Asp97), thereby losing its transcriptional activity [92,93]. Interestingly, caspase-3 is released throughout ROS-induced apoptosis. Briefly, redox imbalance and subsequent accumulation of misfolded proteins result in the perturbation of the endoplasmic reticulum (ER) homeostasis, known as ER stress [94]. Given that the ER functions as a reservoir for calcium (Ca^2+^) and governs intracellular Ca^2+^ homeostasis, any disruption in ER function triggers a swift release of both Ca^2+^ and ROS into the cytosol [95,96,97]. This phenomenon has been observed as a direct outcome of CAP treatment in human cells [98]. Considering that the ER is physically linked to mitochondria, ER stress triggers a surge of Ca^2+^ within mitochondria, resulting in the formation of an apoptosome [99,100,101,102], which in turn, cleaves p65 to inhibit pro-survival NF-κB signalling and ensure cell death. This mitochondria-mediated apoptosis is further accompanied by the oxidation and depolarisation of the mitochondrial membrane, all of which have been demonstrated as direct consequences of CAP treatment in human cells [103,104,105]. 

### 3.4. Interplay of Nrf2 and NF-κB Pathways

Overall, it appears that both the Nrf2/Keap1 and NF-κB/IκBα signalling pathways are involved in CAP’s selective effects. The Nrf2/Keap1 pathway protects non-malignant cells but is compromised in malignant cells due to their redox adaptation limits. Conversely, the NF-κB/IκBα pathway induces selective apoptosis in malignant cells and is suppressed in non-malignant cells. Indeed, there are indications of crosstalk between these pathways [106], for instance, that NF-κB is responsive to oxidative stress by enhancing NF-κB translocation, as well as oxidising cysteine residues in its DNA-binding region [107]. Notably, the reduced state of cysteine residue 62 in the binding region of the p50 subunit is critical for effective DNA-binding by NF-κB, indicating its reliance on several antioxidative enzymes (e.g., Trx and GSH) [108,109,110]. Considering this, the CAP-induced increase in NF-κB translocation might be interpreted differently; diminished enzyme levels in A431 cells and elevated oxidative stress due to CAP exposure might hinder the efficient reduction in NF-κB and thus prevent effective DNA-binding. Consequently, NF-κB may fail to upregulate anti-apoptotic proteins, ultimately leading to apoptosis. Potentially, both mechanisms act synergistically. Furthermore, the NF-κB subunit p65 has been found to counteract the Nrf2/ARE signalling pathway, both directly and indirectly. Specifically, p65 can inhibit Nrf2 by competitively interacting with the transcriptional co-activator CREB binding protein (CBP) and by promoting the recruitment of histone deacetylase 3, a co-repressor, to ARE [111]. Additionally, p65 is also able to directly interact with Keap1 to repress Nrf2 [112]. 

Furthermore, Keap1 represses NF-κB activation by blocking IKKβ phosphorylation and mediating autophagy-dependent IKKβ degradation [113]. Indeed, HaCaT cells displayed elevated Keap1 abundance, whereas IKKβ expression was downregulated (Figure 6). Given the unaltered NF-κB translocation (Figure 5C), it seems plausible that Keap1-mediated IKKβ degradation leads to diminished IκBα phosphorylation/degradation, which, in turn, prevents NF-κB activation. Intriguingly, A431 cells show significantly divergent IKKβ expression relative to their non-malignant counterparts post CAP treatment, implying the absence of Keap1-mediated IKKβ suppression. The divergent inhibition of IKKβ has been reported for other cell lines [114]. However, Keap1 abundance is similar in A431 and HaCaT cells (Figure 6A). Although these findings might seem contradicting, it appears reasonable that elevations in Keap1’s abundance might have completely divergent effects on the cellular systems of both cell lines since Keap1 is known to have more than a dozen binding partners possessing a motif akin to the Nrf2 ETGE motif [115,116]. Thus, Keap1 seems to interact with IKKβ in HaCaT cells, whereas the elevation in Nrf2 abundance in CAP-treated A431 cells (Figure 6A) suggests the involvement of Keap1 in reinforced sequestering of Nrf2. This dynamic finds support in the slight elevation in Nrf2 translocation observed in A431 cells after CAP exposure (Figure 5A). Other studies support the antagonistic nature of NF-κB and Nrf2, reporting a downregulation of gene products controlled by NF-κB, while simultaneously observing an increased expression of genes targeted by Nrf2 as well as Nrf2-mediated repression of NF-κB stimulation [117,118].

### 3.5. Gene Expression in Response to CAP Exposure

Generally, an increase in Nrf2 translocation is ultimately associated with the enhanced activation of genes integral to the oxidative stress response. Consequently, nuclear Nrf2 accumulation is correlated with increased production of antioxidant molecules, including SOD, CAT, Trx, the GSH antioxidant system, and several others [119,120], whereas NF-κB induces the expression of various cytokine genes (e.g., *IL-1*, *IL-6*, *IL-8/CXCL8*, *TNFα,* and *TGF*). We demonstrated significant elevations in the expression of Nrf2-regulated ARE genes *HMOX1* and *GPX1* in non-malignant HaCaT compared to A431 cells. The induction of these genes implies controlled oxidative homeostasis in non-malignant cells, in which we could detect a reinforced translocation of Nrf2 after CAP treatment (Figure 5A). Furthermore, our results show that in malignant A431 cells, the expression of pro-inflammatory genes is downregulated after CAP exposure compared to their non-malignant counterparts (Figure 7). However, indications of immune system activation following CAP treatment have been reported, including the direct activation of immune cells and immunogenic cell death induction in vitro (e.g., in melanoma cells) [71,121]. The expression of *CALR*, a marker for immunogenic cell death, remained unchanged during the given treatment conditions, although induction of immunogenic cell death was demonstrated with other plasma setups. Moreover, we investigated the expression of *SLC7A11,* which was reported to be upregulated 6 h post plasma treatment in tumour cells sensitive towards CAP treatment but not in resistant cells [122]. Unlike their findings, we demonstrated a reduced expression of *SLC7A11* after CAP treatment. Remarkably, unstimulated HaCaT cells also showed a significantly higher *SLC7A11* expression (~2-fold) compared to more sensitive A431 in our experiments. This suggests that the expression of this antiporter is of great importance for the sensitivity to plasma treatment. Also, the expression of *TGFB1* was downregulated after CAP treatment in both cell lines; however, it was more pronounced in malignant A431 cells. This anti-proliferative effect of CAP has been shown in multiple previous studies [21].

Overall, it becomes apparent that cellular responses to CAP are divergent between tumourigenic A431 and non-malignant HaCaT cells. This observation is further supported by a principal component analysis of the gene expression profiles in control and treated A431 and HaCaT cells (Figure 7C). The broad separation of the clusters of all plasma treatment groups reflects the adaptive potential of these cells. In tumour A431 cells, the three CAP-induced patterns cannot be distinctly separated from each other, which is indicative of a low adaptive capability of the cells. However, the clusters of the CAP treatment can be clearly distinguished from the hydrogen peroxide stimulation, which strongly suggests that the plasma impact is multifactorial and not solely based on the generation of oxygen radicals. This highlights the need for future investigations concerning the multifactorial effects of CAP-generated RONS on cells. Indeed, it was demonstrated that CAP treatment generates reactive nitrogen species (RNS) in addition to ROS, which can have both pro- and anti-inflammatory effects depending on the dosage and cell type [15]. Even though it was first assumed that RNS would mainly act by inhibiting phosphotyrosine-dependent signalling owing to their ability to nitrate tyrosine residues within proteins, it later turned out that RNS are also able to upregulate signalling cascades via the inhibition of phosphatases and the direct activation of many different protein kinases. Different studies showed that RNS can act either as an activator of NF-κB signalling via degradation of IκBα [123] or repress NF-κB signalling through nitration on tyrosine residues of p65, leading to its destabilisation, nuclear export, and inactivation of NF-κB activity [124]. Other studies imply an inactivation of IKKβ through exogenous RNS application while simultaneously activating IKKα phosphorylation, suggesting that RNS differentially regulate the classical and the alternative pathways of NF-κB activation. This knowledge implies that RNS might downregulate the expression of pro-inflammatory mediators, thereby providing a counter-regulatory mechanism to prevent over-inflammation in pathological conditions. However, the exact mode of action seems to be dose-dependent and cell-type specific and needs further thorough investigations [125].

## 4. Materials and Methods

### 4.1. Cell Cultivation

A431 and HaCaT cells acquired at CLS (Cell Line Service, Eppelheim, Germany) were used in all experiments. A431 are adherent human SCC, while HaCaT cells are spontaneously transformed human keratinocytes, thus resembling their native phenotype [126]. Both cell lines were cultivated in Dulbecco’s modified Eagle’s media (DMEM; high glucose, GlutaMAX Supplement, 1 mM pyruvate, Invitrogen, Paisley, UK), supplemented with 10% foetal bovine serum (FBS; Sigma-Aldrich, St. Louis, MO, USA) and 1% penicillin/streptomycin (GibcoTM, ThermoFisher Scientific, Waltham, MA, USA) in Cellstar flasks (Greiner Bio-One, Frickenhausen, Germany) and incubated in a humidified 5% (*v*/*v*) CO_2_ atmosphere at 37 °C. In order to detach cells enzymatically, HaCaT cells were incubated with 0.5 mL 0.25% trypsin/EDTA (ThermoFisher Scientific, Waltham, MA, USA) for 7 min while A431 were incubated with 0.5 mL 0.05% trypsin/EDTA for 5 min at 37 °C. For cell counting, the NucleoCounter NC-3000 (ChemoMetec A/S, Steen Søndergaard, Denmark, protocol; “Viability and Cell Count Assay”) was used. For assays, 2 × 10^4^ A431 and HaCaT cells were seeded into a 96-well Cellstar microplate (Greiner Bio-One GmbH, Frickenhausen, Germany), reaching a final volume of 100 μL per well and cultivated for 24 h, unless indicated otherwise.

### 4.2. Plasma Setup and Treatment 

The atmospheric pressure plasma jet kINPen (Neoplas GmbH, Greifswald, Germany; Appendix A) was used for CAP generation in all experiments. This plasma source consists of a quartz capillary with a needle electrode, at which a high-frequency voltage of 2–6 kV at 1.1 MHz is applied. Argon gas (ALPAHGAZ 1, 99,999% purity, AIR LIQUIDE Deutschland GmbH, Germany) served as a feed gas with a flow of 1.9 slm [127,128]. Thus, a low-temperature (<50 °C) plasma jet with a length of 12–14 mm and a width of approx. 1 mm was blown out of the capillary [13]. Unless indicated otherwise, cells and media were directly treated with plasma for 30, 60, 90, 120, 150, or 180 s, respectively. Gas controls were exposed to non-ignited argon gas for 180 s, while untreated samples served as controls. For treatment, the distance from the tip of the plasma jet to the surface of the medium was set to 1 cm. Therefore, the kINPen was fastened in the bracket of a cartesian 2-axis robot (Owis x-y positioning table with phyBASIC precision motor). 

### 4.3. Hydrogen Peroxide Quantification in the Media

The Fluorimetric Hydrogen Peroxide Assay Kit (Sigma-Aldrich, St. Louis, MO, USA) was utilised to quantify the hydrogen peroxide produced in plasma activated media (PAM). For the assay, a hydrogen peroxide standard curve was generated ranging from 0 to 10 µM. At 1 h post treatment, PAM samples were diluted at 1:100. Briefly, 50 µL of each sample and standard was combined with 50 µL of a mastermix comprising peroxidase substrate, peroxidase, and assay buffer. Fluorescence (ex/em: 540/590 nm) was measured immediately with a Tecan Infinite M200 Microplate Reader (Tecan group AG, Männedorf, Switzerland).

### 4.4. MTS Viability Assay 

Following plasma treatment, cells were incubated at 37 °C and 5% CO_2_ for 24 h. Subsequently, 20 μL MTS reagent (Cell Titer 96 Aqueous One Solution Cell Proliferation Assay, Promega, Madison, WI, USA) was pipetted into each well and mixed thoroughly. After 20 min incubation at 37 °C and 5% CO_2_, 80 μL supernatant of each well was transferred into the wells of a new 96-well plate. Absorption (490 nm/reference wavelength 620 nm) measurement was performed with Rosys Anthos Microplate Reader 2012 (Anthos Microsystems GmbH, Friesoythe, Germany). Measured values were offset with blanks and normalised to the controls.

### 4.5. Intracellular Reactive Oxygen Species Quantification

To quantitatively assess basal ROS levels in A431 and HaCaT cells, the DCFDA—Cellular ROS Assay Kit/Reactive Oxygen Species Assay Kit (Abcam, Cambridge, UK) was employed. Therefore, 1 × 10^6^ cells were harvested and resuspended in 999 μL 1 × Buffer and 1 μL of 2′,7′-dichlorofluorescein diacetate (DCFDA). Following 30 min incubation at 37 °C, cells were washed and resuspended in 1 mL phenol-free DMEM devoid of pyruvate (Gibco, ThermoFisher Scientific, Waltham, MA, USA). Thereafter, cells were counted again to ensure an equal cell number. We seeded 2 × 10^4^ stained cells in 100 µL into each well of a 96-well Cellstar microplate (Greiner Bio-One GmbH, Frickenhausen, Germany). Media served as controls, while for positive controls, 0.5 μg/mL hydrogen peroxide was added. Following 2 h incubation, fluorescence (ex/em: 485/535 nm) was detected with a Tecan Infinite M200 Microplate Reader (Tecan group AG, Männedorf, Switzerland).

### 4.6. Quantitative Microscopy 

Prior to cell seeding, cell membranes of HaCaT and A431 cells were labelled utilising a PKH26 Linker-Kit (Sigma-Aldrich, USA). Therefore, 5 × 10^5^ HaCaT and A431 cells were washed in PBS-/- and resuspended in 250 μL Diluent C, respectively. A mixture of 246 μL Diluent C and 4 μL PKH26 dye was added to the cell suspension. Following 10 min incubation at 37 °C, 500 μL pre-warmed serum was added. After 1 min incubation at RT, medium was added. The cell suspension was centrifuged and washed, and subsequently, 5 × 10^4^ cells were seeded into each well of an 8-well chambered coverslip (ibidi GmbH, Germany) filled with 150 μL pre-warmed medium. For positive controls, cells were either treated with 20 μg/mL Poly(I:C) (Invivogen, Toulouse, France) to study the translocation of NF-κB and IκBα or with 15 μg/mL hydrogen peroxide (Merck Schuchardt OHG, Hohenbrunn, Germany) to analyse the translocation of Nrf2 and Keap1. At 1 h post-treatment, cells were washed, fixed with 4% paraformaldehyde (PFA, 10 min, Merck Schuchardt OHG, Germany), permeabilised with 0.01% TritonX-100 (15 min, Sigma-Aldrich Chemie GmbH, Germany), and subsequently, blocked with 5% blocking milk (100 µL, 20 min, Carl Roth GmbH, Karlsruhe, Germany). Then, specimens were incubated with the respective primary antibodies (100 µL, 30 min, RT) for NF-κB/p65, Keap1, pIKK (all Cell Signalling Technology, Danvers, MA, USA), Nrf2 (Abcam, UK), IκBα, and IKKβ (Thermo Fisher Scientific Inc., Waltham, MA, USA). Following two washing steps, cells were incubated with the corresponding secondary antibodies goat anti-rabbit Alexa 488 and goat anti-mouse Alexa 488 (Thermo Fisher Scientific Inc., Waltham, MA, USA, 100 µL, 30 min, RT). Nuclear staining was achieved with Hoechst H33342 (5 µg/mL, 15 min, AppliChem GmbH, Germany). The confocal microscope LSM 780 with ZENblack-Software Version 2.3 SP1 (Carl Zeiss, Oberkochen, Germany) was used for microscopy. All images (1024 × 1024 pixels) used for the analysis were taken with the 20× air objective (EC Plan-Neofluar 20×/0.50 NA). Ten images, each containing approx. 50 to 70 cells, were taken of each specimen. The images were analysed with the R packages readCzi v0.4.1 [129] and cellPixels v0.3.2 [130]. cellPixels identifies nuclei and cytoplasm regions in the images and measures the intensities of specific channels in these regions of interest. Then, intensities per cell were calculated, and the nucleus-to-cytoplasm ratios were formed to analyse the translocation of the targeted proteins (NF-κB, Nrf2, Keap1, IκBα).

### 4.7. Protein Quantification 

For quantification of NF-κB, Nrf2, Keap1, IκBα, IKKβ, and phosphorylated IKK (pIKK), cells were treated with plasma for 180 s. For positive controls, cells were treated with either Poly(I:C) (20 μg/mL; NF-κB, IκBα, IKKβ, pIKK) or hydrogen peroxide (15 μg/mL; Nrf2, Keap1). After 1 h incubation, cells were detached and eight wells of plasma-treated cells were pooled. Subsequently, cells were fixed, permeabilised, and incubated with respective antibodies as described for the quantitative microscopy. After the final washing step, 30 μL of each cell suspension was loaded into the chambers of an NC-Slide A2TM (ChemoMetec A/S, Steen Søndergaard, Denmark). The rendered “FlexiCyte Assay” with a 475 nm LED (em: 560/35 nm) and an exposure time of 2000 ms was selected at the NucleoCounter NC3000TM, and green intensities (protein of interest) were measured to quantify each targeted protein. Data were analysed using the plot manager (NucleoView NC-3000, Version 2.1.25.12).

### 4.8. Quantification of Intracellular Antioxidative Capacity

At 24 h post treatment, cells were lysed using 20 µL of Lysis Buffer (Bio-Rad Laboratories Inc., Hercules, CA, USA) per well, and the protein content was quantified using a Fluorometer (Qubit, ThermoFisher Scientific, Waltham, MA, USA). For evaluation of the cellular antioxidative capacity, the Antioxidant Assay Kit (Cayman Chemical, Ann Arbor, MI, USA) was used according to the manufacturer’s instructions. Briefly, 10 µL of metmyoglobin and 150 µL of chromogen were added to each sample containing 10 µg protein and a Trolox standard ranging from 0 to 0.495 mM. The reaction was initiated with the addition of 40 µL of hydrogen peroxide to each well. Absorbance (750 nm) was measured in duplicate using a Tecan Infinite M200 Microplate Reader (Tecan group AG, Männedorf, Switzerland).

### 4.9. Gene Expression Analysis

Cells were exposed to plasma for 60, 90, and 120 s, gas controls were subjected to argon gas for 120 s, and positive controls were treated with 15 µg/mL hydrogen peroxide and 20 µg/mL Poly(I:C), respectively, for 6 h. Subsequently, 50 µL RTL Buffer (Qiagen, Hilden, Germany) was added for cell lysis. Then, total RNA was isolated from both cell lines using the ISOLATE II RNA Mini Kit (Meridian Bioscience Inc., Cincinnati, OH, USA), according to the manufacturer’s instructions. The extracted RNA was measured with a NanoDrop OneC (ThermoFisher Scientific, Waltham, MA, USA) to determine its concentration and purity and then transcribed into cDNA using the SensiFAST cDNA Synthesis Kit (Bioline/Meridian Bioscience Inc., Cincinnati, OH, USA). The protocol for the subsequent quantitative PCR (qPCR) was optimised for a 12 μL reaction volume using 6 μL SensiFAST SYBR No-ROX Mix (Bioline/Meridian Bioscience Inc., Cincinnati, OH, USA), 5 μL cDNA, and 1 μL primers (sense and antisense). The pyrosequencing assay design software (Biotage AB, Uppsala, Sweden) was used to design the sense and antisense primers (Table 1) for each of the nine target genes and two of the three reference genes, comprising *YWHAH* [131], *KRT14*, and *RPL13A*. The proper function of each primer pair was validated via standard PCR; in addition, amplicon-specific standard curves (based on 107–103 copies per 5 μL) were generated (R2 > 0.999) to assess the PCR efficiency.

The qPCRs were conducted in a LightCycler-96 instrument (Roche); the program included an initial denaturation (95 °C, 5 min), followed by 40 cycles of denaturation (95 °C, 5 s), annealing (60 °C, 15 s), and elongation (72 °C, 15 s), as well as the fluorescence measurement (72 °C, 10 s). No-template controls were included to monitor contaminations. Amplicons were visualised on agarose gels in order to assess their product size and quality. In addition, all melting curves generated were analysed individually to validate the absence of unspecific products. Quantification cycles (Cq) between 5 and 35 were considered relevant for further analysis.

### 4.10. Assessment of Cellular Oxygen Consumption 

Cells were treated with CAP for 60 and 90 s as well as 15 µg/mL hydrogen peroxide for positive controls (=441 µM). Directly after treatment, the Resipher (Lucid Scientific Inc., Atlanta, GA, USA) system (a lid with attached sensors) was placed on the plate, and oxygen consumption was monitored continuously over 24 h at 37 °C and 5% CO_2_. Due to an equilibration phase for stabilisation of the temperature and oxygen content in the incubator, values starting at 3 h after treatment were analysed. Measured values were normalised to controls. 

### 4.11. Statistical Analysis 

Unless stated otherwise, the presented data are expressed as the mean ± standard deviation or box–whisker plots, calculated from a minimum of three independent experimental replicates (n ≥ 3). Statistical analyses were conducted utilising GraphPad Prism 8 software (GraphPad Software, San Diego, CA, USA). For datasets containing more than 20 data points, the normality of the distribution was assessed by employing the Kolmogorov–Smirnov test. Based on the distribution, the unpaired *t*-test or Kruskal–Wallis test for non-parametric datasets was applied, followed by the Tukey test to detect significant intergroup variations. Statistical significance was assumed at a probability of error (*p*-value) of ≤0.05 and indicated as follows: * *p* ≤ 0.05; ** *p* ≤ 0.01; *** *p* ≤ 0.001; **** *p* ≤ 0.0001. The Cq values of the quantified transcripts were translated into copy numbers based on external amplicon-specific standard curves (as described above). The obtained copies were all normalised with a factor based on the geometric mean values from the reference genes *YWHAH*, *RPL13A,* and *KRT14*.

## 5. Conclusions

The results obtained in this work indicate that CAP treatment selectively activates NF-κB in A431 cells in a dose-dependent manner. This is potentially triggered by the increasing oxidative stress, pushing cells to surpass their apoptotic threshold. Nrf2 and NF-κB exhibit several points of intersection that indicate their antagonistic relationship. This is supported by the opposing translocation patterns of Nrf2 and NF-κB between A431 and HaCaT cells. In CAP-exposed HaCaT cells, Nrf2 translocation is enhanced to enable redox adaptation, while NF-κB signalling is inhibited. Conversely, A431 cells exhibit increased basal nuclear Nrf2 localisation as an adaptation to intrinsic ROS elevation, allowing only minimal enhancement after CAP exposure. Thus, the burdened antioxidative system in A431 cells is overwhelmed, resulting in the activation of the NF-κB signalling pathway leading to apoptosis. 

Studies have shown that ROS-inducing therapies are an effective anti-cancer strategy by increasing ROS levels. Consequently, the effect of CAP on cellular ROS levels and key players of oxidative response should be further explored and optimised to enhance its effectiveness in tumour cells while minimising side effects on non-malignant cells. In essence, the outcomes of this work highlight the significance of incorporating non-malignant counterparts in future investigations. Such inclusion would allow a more profound understanding of the intricate mechanisms underlying CAP’s selectivity towards malignant cells. For instance, a deeper understanding of NF-κB’s role in respective cancer types would allow us to optimise the therapeutic effectiveness and reduce adverse effects. Also, the combination of CAP with other ROS-inducing therapies or chemotherapeutic agents could lead to new strategies for treating resistant cancers [132,133].

Furthermore, it is crucial to recognise that while much emphasis has been placed on the study of ROS, RNS have also been demonstrated to be essential [134,135]. The double-edged effects of RNS on both signalling pathways and the time-dependent interaction of involved proteins and their translocation require more in-depth investigation. Thus, forthcoming studies should elucidate the primary mediators of CAP’s effects, as well as the functions of individual reactive species [136].

Even though the outcomes of initial clinical trials targeting head and neck SCC show signs of the effectiveness and safety of CAP [137], the untapped potential that a deep comprehension of the definite CAP-triggered intracellular mechanisms harbours should not be underestimated. The possible involvement of other signalling molecules, also considering metabolic adaptation of CAP-treated cells, requires further investigation. Such metabolic adaptation could be facilitated by the HIF1α (Hypoxia Inducible Factor 1 Subunit Alpha) TF involved in stress-response mechanisms, which may induce resistance in solid tumours [138]. Thus, it would be interesting to investigate this hypothesis in future studies. Generally, it should be acknowledged that a profound understanding of the underlying processes of CAP is fundamental to unveiling CAP’s full potential in secure cancer therapy. 

## Figures and Tables

**Figure 1 ijms-25-10967-f001:**
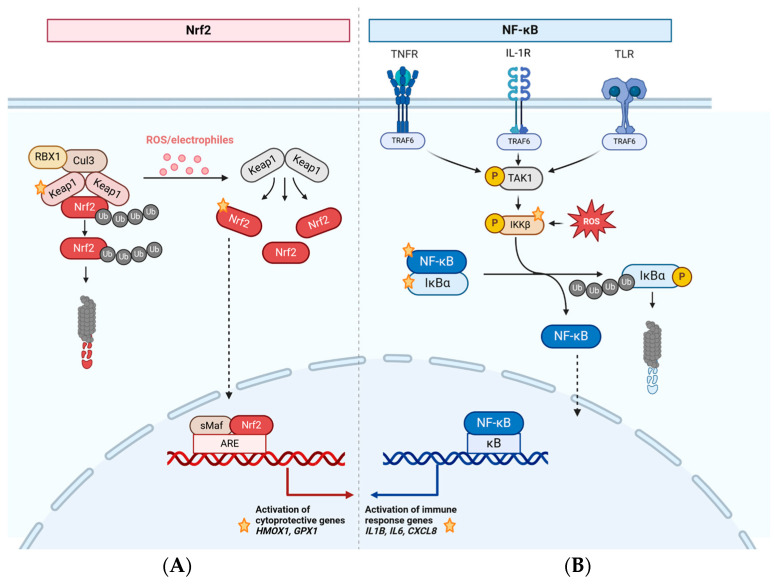
The Nrf2/Keap1/ARE and NF-κB/IκB signalling pathways. Proteins/genes investigated in this study are marked with a star. (**A**) Under basal conditions, Keap1 molecules bind Nrf2, facilitating its labelling with ubiquitin and targeting it for proteasomal degradation, while Keap1 is regenerated. During oxidative stress, the Keap1 cysteine thiols react to a broad range of oxidative and electrophile molecules leading to conformational changes interrupting the interactions between Nrf2 and Keap1. Thus, Nrf2 is released to translocate into the nucleus, where it binds to the ARE present in the promotors of numerous cytoprotective genes. (**B**) Regulation of NF-κB signalling occurs via activation of the trimeric kinase complex (IKKα, IKKβ, IKKγ) via the upstream TAK1. IKK-mediated IκBα phosphorylation is followed by IκBα degradation and rapid nuclear translocation of NF-κB, where it activates genes for immune regulation and apoptosis pathways. [ARE: antioxidant response element, Cul3: Cullin 3, *CXCL8*: chemokine (C-X-C motif) ligand 8, *GPX1*: glutathione peroxidase 1, *HMOX1*: heme oxygenase 1, IκBα: inhibitor of κBα, IKKβ: IκB kinase β, IL-1R: interleukin 1 receptor, NF-κB: nuclear factor-kappa B, Nrf2: nuclear factor-erythroid 2-related factor 2, P: phosphate, RBX1: RING-box protein 1, ROS: reactive oxygen species, sMaf: small Maf, TLR: toll-like receptor, TNFR: tumour necrosis factor receptor, TRAF6: TNF receptor-associated factor 6, Ub: ubiquitin] (created in BioRender. Rebl, H. (2024) BioRender.com/s71v052 accessed on 1 October 2024).

**Figure 2 ijms-25-10967-f002:**
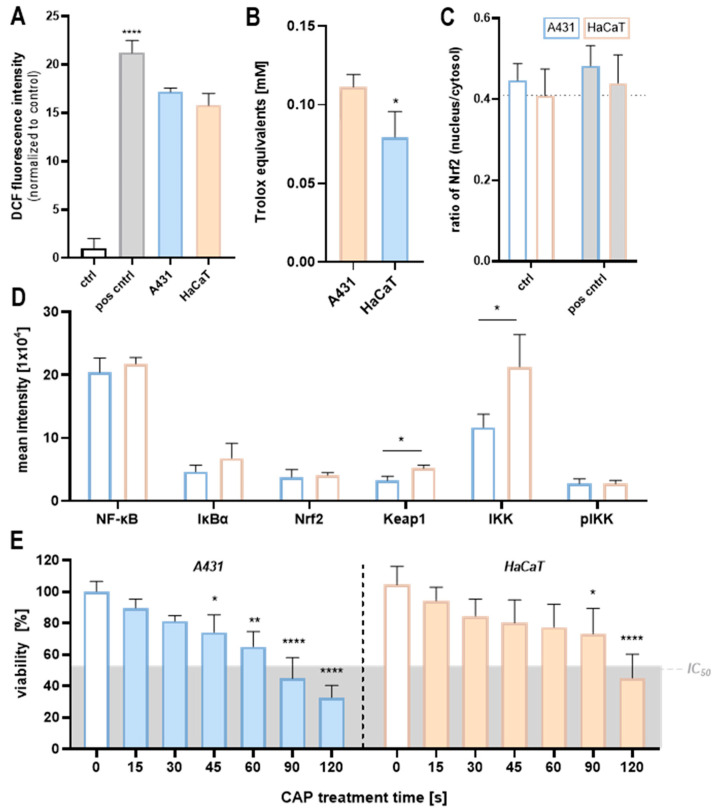
Characterisation of unstimulated dermal cells. (**A**) Intracellular levels of reactive oxygen species (ROS) are comparable in both cell lines (n = 2), where 0.5 µg/mL H_2_O_2_ served as a positive control. (**B**) The basal cellular antioxidant level (displayed as Trolox equivalents) is significantly lower in HaCaT (orange) cells relative to A431 (blue) cells (n = 4). (**C**) A431 cells show a slightly higher nuclear/cytosolic ratio of Nrf2 in a basal state compared to HaCaT cells (dotted line indicates the basal level of HaCaT; n = 3), where 15 µg/mL H_2_O_2_ served as positive control. (**D**) Protein abundances of NF-κB, IκBα, Nrf2, Keap1, IKK, and pIKK (n = 3). HaCaT cells show significantly higher protein levels of Keap1 and IKK compared to A431 cells. (**E**) Cell viability of A431 cells significantly and dose-dependently decreases after a CAP treatment durations of ≥45 s, whereas HaCaT cells display a slower decline with significance appearing after ≥90 s of treatment (n = 3). * Compared to the respective control (0 s) (* *p* ≤ 0.05; ** *p* ≤ 0.01; **** *p* ≤ 0.0001).

**Figure 3 ijms-25-10967-f003:**
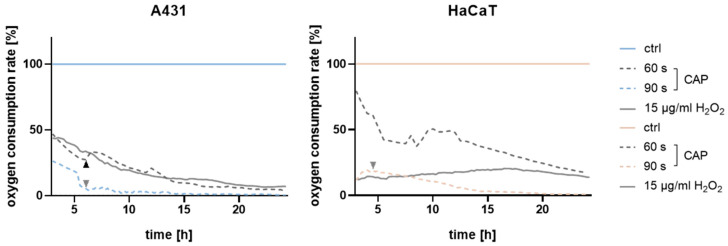
Oxygen consumption rates of A431 (**left**) and HaCaT (**right**) cells after treatment with cold atmospheric pressure plasma (CAP). Values starting at 3 h after treatment were analysed, and unstimulated cells were normalised to a value of 1. CAP treatment of 60 s led to a significant reduction (black arrow) from tA= 5.5 h only for A431, whereas 90 s exposure showed a significant decrease (grey arrows) from tA= 5.5 h (A431) and tH = 4.5 h (HaCaT) (lines show mean values from n = 4). Arrows mark the timepoints where alterations in the oxygen consumption rate start to be significant compared to the respective untreated control (0 s). Statistical analyses were performed using multiple *t*-tests, and statistical significance was determined with the Holm–Sidak method (alpha = 0.05).

**Figure 4 ijms-25-10967-f004:**
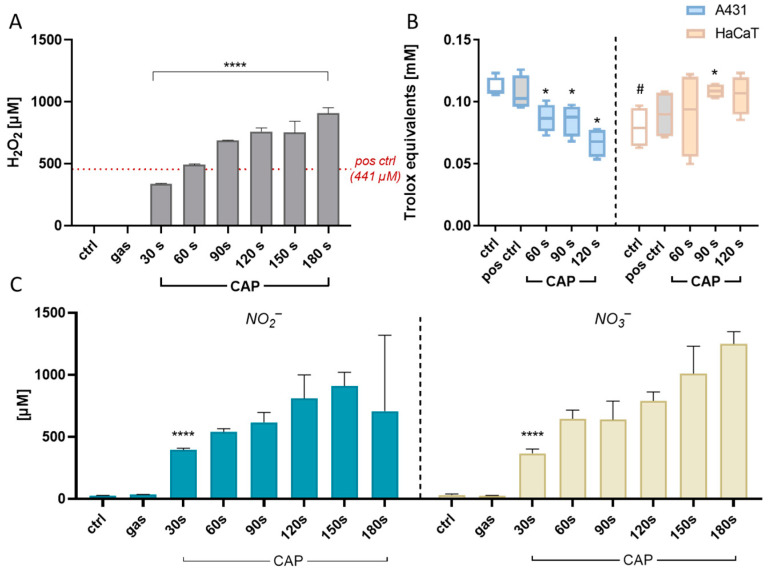
(**A**) Measurement of ROS in media treated with cold atmospheric pressure plasma (CAP). Hydrogen peroxide (H_2_O_2_) concentrations significantly increase in plasma-activated media in a dose-dependent manner. The H_2_O_2_ concentration used as a positive control in all experiments (441 µM, red dotted line) is nearly equivalent to the hydrogen peroxide content in 60 s CAP treatment (492 µM) (n = 3). (**B**) Antioxidative capacity displayed as Trolox equivalents after CAP exposure. The antioxidative capacity of untreated tumour cells is significantly higher than in HaCaT cells; however, the antioxidant level drops in these cells post CAP treatment. In contrast, plasma treatment leads to elevated antioxidant levels in HaCaT cells. p # compared to A431 negative control, * compared to the respective negative control (n = 4). (**C**) Determination of RNS in media after CAP treatment. Nitrite (left, blue, NO_2_^−^) and nitrate (right, beige, NO_3_^−^) concentrations significantly increase with a longer treatment duration (**** *p* ≤ 0.0001, n = 3).

**Figure 5 ijms-25-10967-f005:**
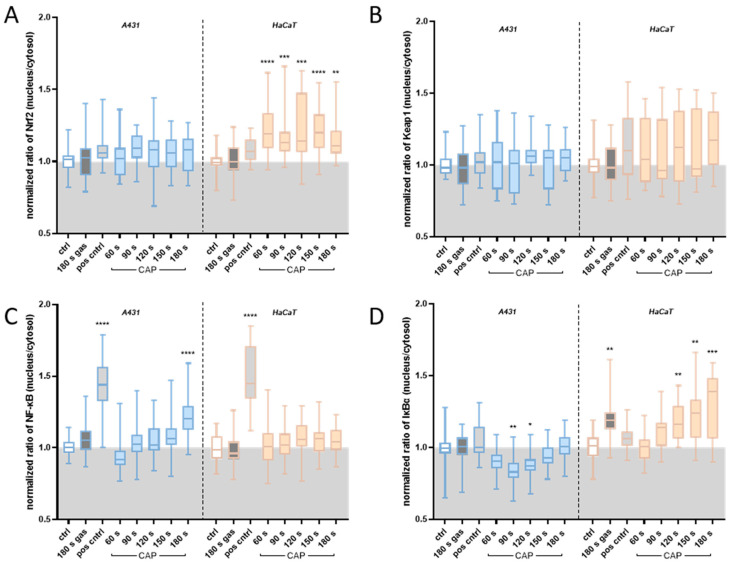
Translocation dynamics of Nrf2 (**A**), Keap1 (**B**), NF-κB (**C**), and IκBα (**D**) in A431 (blue) and HaCaT (orange) cells after cold atmospheric pressure plasma (CAP) exposure. (**A**) In HaCaT cells, Nrf2 translocation is significantly increased after CAP treatment—independent of the dose, whereas A431 cells show a marginal upward trend (n = 3). (**B**) No significant changes in the Keap1 localisation are apparent in either cell line (n = 3). (**C**) In A431 cells, NF-κB translocation is significantly increased after 180 s of CAP treatment, whereas HaCaT cells show no noteworthy alterations (n = 4). (**D**) Generally, IκBα translocation increases dose-dependently in both cell lines. However, A431 cells display a significant decrease after 90 and 120 s of CAP treatment, gradually giving way to an increase with an escalating exposure time, ultimately converging to the control (white filling) ratio of 1 after 180 s. Dark grey filling: treatment with argon gas only; light grey filling: positive ctrl; (* *p* ≤ 0.05; ** *p* ≤ 0.01; *** *p* ≤ 0.001; **** *p* ≤ 0.0001, n = 3).

**Figure 7 ijms-25-10967-f007:**
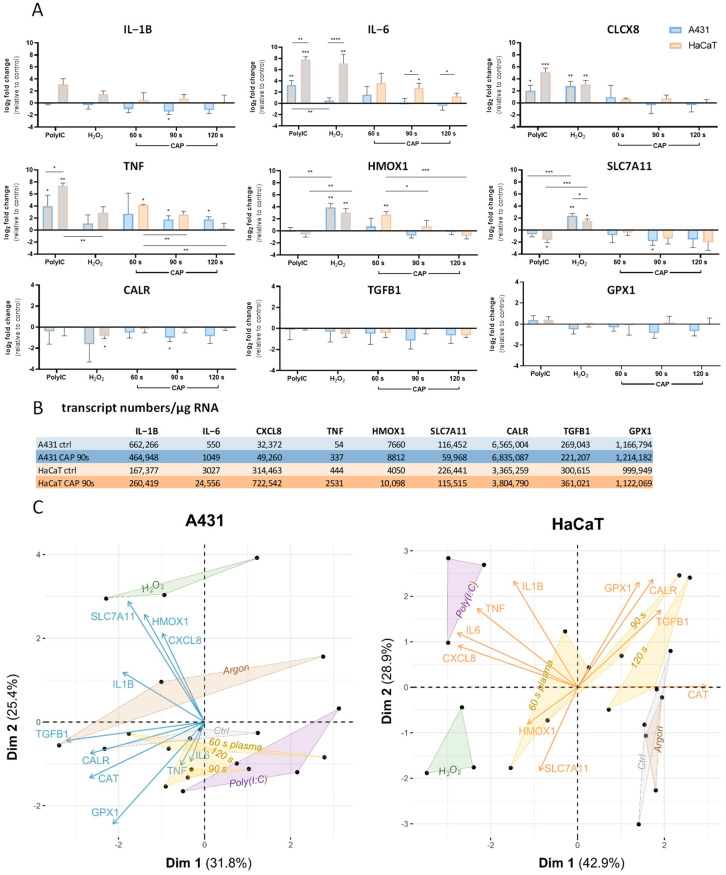
Expression analyses of Nrf2- and NF-κB -associated genes in A431 and HaCaT cells treated with cold atmospheric pressure plasma (CAP). (**A**) Log2-fold change levels of the indicated genes relative to the respective untreated control. Positive controls are indicated in grey. CAP leads to an activation of *IL-6*, *TNF*, and *HMOX* expression in HaCaT, while A431 expression levels are moderately affected (* *p* ≤ 0.05; ** *p* ≤ 0.01; *** *p* ≤ 0.001; **** *p* ≤ 0.0001). (**B**) Transcript numbers/µg RNA show the low copy genes *TNF*, *HMOX1*, and *IL-6* in both unstimulated dermal cell lines, while the other genes examined are referred to as high copy genes. (**C**) The factor maps indicate the contribution of individual genes to the overall separation of log10-transformed expression data in A431 and HaCaT cells. The lengths of the blue and orange vectors radiating from the centroid (0.0) indicate the contribution of each gene to the overall variance of data in A431 and HaCaT cells, respectively. The majority of variance is described by the dimensions Dim1 and Dim2 given in parentheses on the axis labelling. The coloured clusters (ctrl, grey; plasma, yellow; argon, brown; H_2_O_2_, green; poly[I:C], purple) indicate samples treated in the same way.

**Table 1 ijms-25-10967-t001:** Primers used in this study.

Gene Symbol	Gene Product	Function	NCBI Accession Code	Sense and Antisense Primer (5′-3′)	Primer Efficiency [%]	Amplicon Length [bp]
** Target genes: **						
*CALR*	Calreticulin	Calcium homeostasis	NM_004343	CAGGTCAAGTCTGGCACCATC, GCGTAACAAAGGCAGCAGAGAA	89.5	107
*CXCL8*	C-X-C motif chemokine ligand 8 (interleukin-8)	Chemotaxis and immunoregulation	NM_000584	AGATGTCAGTGCATAAAGACATAC, TCTGTCTGGACCCCAAGGAAAA	101.7	152
*GPX1*	Glutathione peroxidase 1	Protecting from oxidative damage	NM_000581	ATCAGGAGAACGCCAAGAACGA, TCATGCTCTTCGAGAAGTGCGA	102.3	100
*HMOX1*	Heme oxygenase 1	Protection from oxidative damage	NM_002133	CGTTCCTGCTCAACATCCAGCT, GATTCTGCCCCCGTGGAGAC	97.1	139
*IL1B*	Interleukin-1 beta	Inflammation	NM_000576	GTACAAGGAGAAGAAAGTAATGAC, CAAAGAAGAAGATGGAAAAGCGAT	112.1	157
*IL6*	Interleukin-6	Inflammation	NM_000600	AGTAACATGTGTGAAAGCAGCAAA, TGGTCTTTTGGAGTTTGAGGTATA	102.0	152
*SLC7A11*	Solute carrier family 7 member 11	Glutathione production	NM_014331	CAAGGTGCCACTGTTCATCCC, TACAGGGATTGGCTTCGTCATC	102.2	106
*TGFB1*	Transforming growth factor beta 1	Control of cell growth, proliferation and differentiation	NM_000660	AGCACGTGGAGCTGTACCAGA, CCTTAGCGCCCACTGCTCCT	98.7	177
*TNF*	Tumour necrosis factor alpha	Inflammation	NM_000594	GGCCCCCAGAGGGAAGAGTT, AGTCAGATCATCTTCTCGAACCC	99.0	82
** Reference genes: **						
*KRT14*	Keratin 14	Cytoskeleton formation	NM_000526	GAGGAGATGAATGCCCTGAGAG, AAGGATGCCGAGGAATGGTTCT	N/A	157
*RPL13A*	Ribosomal protein L13a	Protein synthesis	NM_012423	CAAGCGGATGAACACCAACCC, CCACAAAACCAAGCGAGGCCA	N/A	111

## Data Availability

The data that support the findings of this study can be found in the Zenodo repository at 10.5281/zenodo.13864976.

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
