# Peer review of "Interplay of Cellular Nrf2/NF-κB Signalling after Plasma Stimulation of Malignant vs. Non-Malignant Dermal Cells"

_ijms, 2024, doi:10.3390/ijms252010967_

Round 1

Reviewer 1 Report

Comments and Suggestions for Authors

The manuscript “Interplay of Cellular Nrf2/NFκB Signalling after Plasma Stimulation of Malignant vs. Non-Malignant Dermal Cells” describes in-depth analyses of the antioxidant pathways which aim to elucidate the molecular basis for differences in responses to cold plasma using skin cell lines A431 and HaCaT. The results presented here support the hypothesis outlined that malignant cells may need to counteract higher basal ROS levels which limits their adaptability to additional oxidative stress.

The manuscript is well-written and results are well-presented. The introduction provides a detailed explanation of the Nrf2 and NFκB pathways and helps to understand the hypothesis. The methodology is  described in detail. I commend the authors on this in-depth mechanistic study and the range of analyses employed and believe that this can be an important contribution to our understanding of CAP-induced responses.

Nonetheless, I have a few comments/questions which I think should be addressed:

-          Why was the PAM incubated or 1 hour prior to hydrogen peroxide quantification?  (some reactive species may react with media components during this time?!)

-          Fig. 2: it would be useful to indicate in the legend which concentration of H2O2 was used as pos. control

-          Figure 3: I suggest indicating in the legend that measurements from 3h post treatment are shown in the graph

-          Fig. 3: The legend mentions ‘significant reduction’ which suggest statistical analysis. Please provide details on the analysis performed here (compared to control?). N=4 is indicated but no error bars are shown in the graph. Why is the downward trend in HaCaT insignificant and is this the case for both 60 and 90 sec? (There seems to be a big difference between the control and the treated cells.)

-          Fig. 3: It is not clear from looking at the graph why the time points marked with arrows were selected. The interpretation of Fig. 3 seems very short and there is limited interpretation of the data being shown. Oxygen consumption in A431 seems to be reduced to 0 until 5h and then slowly starts to increase a little bit? What do the authors propose is happening here?

-          Viability data in Fig. 2 suggest 40 and 60% of cells remain viable after 60 and 90sec treatment in A431 and >70% in HaCaT cells yet there is almost no oxygen consumption for either cell line after 90 sec treatment – how do you explain this discrepancy?

-          HaCat cells show a very different response to H2O2 in terms of oxygen consumption – do the authors have any possible explanation for this? In general I think the data on oxygen consumption needs to be described and explained in more detail.

-          Fig. 4B: it would be beneficial if the positive control (H2O2) was included here.

-          Fig. 6: it would be helpful to specify the concentration of H2O2 used as positive control in the legend. It was mentioned elsewhere that 441uM H2O2 was used which corresponds roughly to 60sec plasma treatment. Here 180sec of CAP are used. Would a higher concentration of H2O2 have been a more suitable control? The figure legend states that “In A431 cells Keap1 abundance was significantly increased after 180 s exposure to cold atmospheric pressure plasma (CAP) (A, right).” However, no significance is indicated in the graph.

-          The authors mention that the spontaneously transformed HaCaT cells retain the original cell phenotype. I would nonetheless be careful with regards to describing these as ‘healthy’ cells. I think the distinction malignant vs non-malignant is the more accurate. While HaCaT cells are not malignant, they are nonetheless not ‘normal’ cells as they are immortalized and hence possess features which a ‘normal’ skin cell would not have. My advice would be to avoid referring to these as healthy and also to discuss this aspect as a potential limitation of the conclusion which can be drawn from the comparison of the 2 cell lines.

-          Section 2.2 is entitled “Selective inhibitory effect of CAP on dermal tumour cells.” My advice would be to change this heading to sth like” comparison of CAP effect between…” . Different cell lines exhibit differences in sensitivity to CAP treatment not solely based on whether they are tumour cells or not. Hence I feel that only a difference in viability/metabolic activity post treatment is possibly not sufficient to conclude a selective effect. Likewise I would advise more cautious phrasing of “It is well acknowledged that CAP exhibits a selective inhibitory effect on tumour cells.” (e.g. “several studies have shown that CAP exhibits a selective inhibitory effect on tumour cells (ref)”) as there are also other studies which were not able to determine such selectivity and there may be cell-specific differences.

-          Have the authors compared apoptosis between the 2 cell lines (using AnnexinV, caspase activation or similar)? They conclude that “Ultimately, the inability of cancer cells to adapt to additional ROS as inflicted by CAP leads to ROS over-loading and the accelerated surpassing of the threshold, resulting in apoptosis of A431 cells.” And “Thus, the burdened antioxidative system in A431 cells is overwhelmed, resulting in the activation of the NFκB signalling pathway leading to apoptosis” While this does seem like a plausible conclusion, in the absence of data showing that apoptosis is actually induced, an important piece of the overall picture is missing.

-          Page 13, line 386: “In addition to the stronger effect on the respiration of tumour cells, earlier experiments demonstrated that CAP also significantly impairs the adhesion capacity of malignant cells to a greater extent than their healthy counterparts.” It is not clear how this is linked to cell respiration and reference to this study seems a bit out of place.

Formatting and language        

-          Figures should ideally be arranged in a way that the text presenting/describing the results comes before the figure (e.g. Fig 3 and it’s corresponding text) to make it easier for the reader

-          Cell concentrations should be superscript, e.g.  “For assays, 2 × 104 A431,..” the 4 should be superscript

-          Page 3, line 108 refers to Figure S1 but I believe this should be Figure 1

Author Response

Thank you for your valuable recommendations.

please refer to the attached document

Reviewer 2 Report

Comments and Suggestions for Authors

Overview

The manuscript explores the interplay between Nrf2 and NFκB signaling pathways in malignant (A431) vs. non-malignant (HaCaT) dermal cells when exposed to Cold Atmospheric Pressure Plasma (CAP). The study demonstrates that CAP selectively induces apoptosis in cancer cells (A431) by disrupting their redox balance, while non-malignant cells (HaCaT) successfully adapt by activating Nrf2 signaling. This differential response offers insights into how CAP could be used in targeted cancer therapies by leveraging the vulnerabilities of cancer cells to oxidative stress.

Overall Structure and Flow

The manuscript is generally well-structured, with clear delineation between the introduction, results, and discussion sections. However, certain areas can benefit from more explicit transitions between sections to enhance readability and guide the reader smoothly through your argumentation.

Suggestion: Consider adding transition sentences at the end of each major section to link it to the next. For example, after the section on basal redox adaptation, a phrase like “Having established the differences in basal redox status, we next explore how these variations modulate cellular responses to CAP-induced oxidative stress.”

Introduction

The introduction does an excellent job of contextualizing skin cancer's global burden and Cold Atmospheric Pressure Plasma (CAP) as an emerging treatment modality. However, some elements, such as the mechanisms behind CAP and the importance of the Nrf2/NFκB pathways, could be more clearly emphasized.

Clarification: You mention that CAP induces a “selective effect” on cancer cells, but you may want to delve deeper into the hypothesized molecular mechanisms by which CAP selectively affects cancerous vs. non-cancerous cells.

References: The use of references [1-4] to establish CAP's efficacy could be expanded. More up-to-date studies might strengthen the introductory rationale for choosing CAP as a treatment model.

Results

The experimental data are presented thoroughly, with clear distinctions between malignant A431 and non-malignant HaCaT cells regarding ROS levels, antioxidant responses, and signaling pathways.

a.     Figures

The figures provided (e.g., NFκB and Nrf2 translocation dynamics) are informative, but the labeling could be more descriptive, especially for readers less familiar with the methodology.

Suggestion: Enhance figure legends with a more detailed description. For example, in Figure 5, describe what the different colors of bars represent (e.g., untreated vs. CAP-treated), and briefly mention why translocation dynamics are crucial in understanding the pathways.

b.     Protein Quantification

The comparative protein quantification of NFκB, IκBα, Nrf2, Keap1, and others is solid. However, it would be helpful to explain why some proteins (e.g., IKKβ) showed unexpected patterns of expression, particularly in A431 cells.

Additional Insight: Consider a more detailed hypothesis on the role of IKKβ and why its regulation might diverge between malignant and non-malignant cells, especially under CAP exposure.

Discussion

The discussion provides a comprehensive analysis of your findings, focusing on the differential CAP response in cancerous vs. healthy cells. However, certain points would benefit from deeper discussion or alternative interpretations.

a.     Oxidative Stress Response

The discussion on the heightened oxidative stress in A431 cells is intriguing. You suggest that the basal redox adaptation in these cells could explain their inability to cope with additional CAP-induced stress. While this is valid, consider expanding the discussion around Nrf2’s role in cancer cells’ resistance mechanisms, which is a critical area of cancer biology research.

Clarification: You might expand on how Nrf2’s regulatory role becomes dysregulated in cancer cells, contributing to both resistance to therapy and the elevated antioxidant capacity seen in your data.

b.     NFκB Activation

Your results on NFκB’s differential role—pro-apoptotic in A431 cells vs. non-activation in HaCaT cells—could be more explicitly tied to clinical relevance. Given NFκB’s known role in inflammation and apoptosis, it would be helpful to speculate on how targeting this pathway could enhance CAP therapy's effectiveness.

Insight: NFκB is traditionally associated with inflammation and cell survival, but your data suggest an apoptotic role in A431 cells. Discuss how this dual role can be leveraged therapeutically in targeting resistant cancers.

Language and Terminology

In general, the language is formal and appropriate for an academic manuscript, though there are areas where clarity could be improved.

Ambiguity: In some sentences, technical jargon can obscure meaning. For example, “CAP treatment significantly elevated Nrf2 nuclear translocation in non-malignant HaCaT cells, indicating a strong antioxidant response.” You could clarify what is implied by “strong antioxidant response” in terms of therapeutic potential or oxidative stress mitigation.

Grammar: Minor grammatical improvements are suggested for conciseness, such as avoiding redundancy in sentences like “This notion finds support in the significantly higher antioxidant levels assessed in unstimulated A431 cells compared to their non-malignant counterparts.” Suggestion: "This notion is supported by the higher antioxidant levels in unstimulated A431 cells."

Conclusion

The conclusion ties the findings back to the broader context of CAP’s therapeutic potential. However, it could be stronger by highlighting potential next steps or unanswered questions in your research, such as further studies needed to confirm the role of NFκB in CAP-induced apoptosis.

Future Directions: Speculate on how this research could translate into clinical trials or combination therapies with other oxidative stress-inducing agents.

Final Recommendations:

·       Clarify key terms for broader accessibility.

·       Provide a more in-depth explanation of anomalous protein expression patterns.

·        Strengthen the discussion on clinical implications, especially in terms of redox homeostasis and apoptotic signaling pathways.

Author Response

(The authors gave the same response as above.)
